# Effect of Si Doping on the Performance of GaN Schottky Barrier Ultraviolet Photodetector Grown on Si Substrate

**Fangzhou Liang** [1,2]**, Wen Chen** [1,3]**, Meixin Feng** [1,3,*]**, Yingnan Huang** [1,3]**, Jianxun Liu** [1,4]**, Xiujian Sun** [1,4]**, Xiaoning Zhan** [1,4]**, Qian Sun** [1,3,4]**, Qibao Wu** [5] **and Hui Yang** [1,3,4]

1    Key Laboratory of Nano-Devices and Applications, Suzhou Institute of Nano-Tech and Nano-Bionics (SINANO), Chinese Academy of Sciences (CAS), Suzhou 215123, China; fzliang2017@sinano.ac.cn (F.L.); wenchen2020@sinano.ac.cn (W.C.); ynhuang2018@sinano.ac.cn (Y.H.); jxliu2018@sinano.ac.cn (J.L.); xjsun2018@sinano.ac.cn (X.S.); xnzhan2018@sinano.ac.cn (X.Z.); qsun2011@sinano.ac.cn (Q.S.); hyang2006@sinano.ac.cn (H.Y.)
2    School of Microelectronic, University of Science and Technology of China (USTC), Hefei 230026, China
3    School of Nano Technology and Nano Bionics, USTC, Hefei 230026, China
4    Guangdong (Foshan) Branch, SINANO, CAS, Foshan 528000, China
5    School of Intelligent Manufacturing and Equipment, Shenzhen Institute of Information Technology, Shenzhen 518172, China; wuqb@sziit.edu.cn
*    Correspondence: mxfeng2011@sinano.ac.cn

**Abstract:** GaN Schottky barrier ultraviolet photodetectors with unintentionally doped GaN and lightly Si-doped $n^{-}$-GaN absorption layers were successfully fabricated, respectively. The high-quality GaN films on the Si substrate both have a fairly low dislocation density and point defect concentration. More importantly, the effect of Si doping on the performance of the GaN-on-Si Schottky barrier ultraviolet photodetector was studied. It was found that light Si doping in the absorption layer can significantly increase the responsivity under reverse bias, which might be attributed to the persistent photoconductivity that originates from the lowering of the Schottky barrier height. In addition, the devices with unintentionally doped GaN demonstrated a relatively high-speed photo response. We briefly studied the mechanism of changes in Schottky barrier, dark current and the characteristic of response time.

**Keywords:** GaN on Si; ultraviolet photodetector; Si doping; responsivity; persistent photoconductivity





## 1. Introduction

GaN, with a natural direct bandgap of approximately 3.4 eV, has a very high absorption coefficient for ultraviolet (UV) light and has become an ideal semiconductor material used for UV detection [1,2]. GaN-based UV photodetectors (PDs) have a small size and high efficiency and can be widely used in the civilian and military field, such as in biological sensing, flame monitoring and missile plume detection.

Normally, GaN-based UV PDs are grown on a conventional sapphire substrate. Compared with a sapphire substrate, a Si substrate has a larger wafer size and lower material cost. More importantly, a Si substrate is thermally matched with a Si-based read-out integrated circuit (ROIC), which can significantly improve the device's reliability while integrated with the Si-based ROIC [3,4]. Thus, fabrication of GaN-based UV PDs on a Si substrate can both reduce the device cost and enhance the reliability, qualities which are becoming increasingly important nowadays.

However, there are few studies on GaN-on-Si UV PDs in the existing literature due to the huge lattice and thermal mismatches between GaN and Si, which usually result in a high threading dislocation (TD) density, large tensile stress, wafer bowing and even micro-crack networks, severely affecting the device's performance and reliability. In addition to these factors, intentional Si doping in the absorption layer also has a significant impact on

the device's performance, such as the reverse leakage current, responsivity and response time [5]. There are few studies on the effect of Si doping in GaN UV PDs.

In this study, high-quality GaN-on-Si materials were produced, and then high-performance GaN-on-Si Schottky barrier UV PDs were fabricated. After this, we studied the influence of Si doping on the electrical performance, spectral response and response time of the GaN-on-Si UV PDs.

## 2. Materials and Methods

Samples A and B have the same epitaxial structure and were grown on Si(111) substrates by metal–organic chemical vapor deposition (MOCVD). Ammonia ($NH_3$), trimethylgallium (TMGa) and silane ($SiH_4$) were used as N, Ga and Si precursors, respectively. Figure 1a shows a structural sketch of samples A and B. An Al-composition graded AlN/AlGaN buffer layer was firstly grown on the Si substrates to engineer the tensile stress and filter the TDs [4,6,7], and then a 1-μm-thick Si-doped n-GaN layer (n ≈ $5 \times 10^{18}$ cm$^{-3}$) was developed for both samples, followed by a 3-μm-thick unintentionally doped (UID) GaN layer (u-GaN, n ≈ $2 \times 10^{15}$ cm$^{-3}$) for sample A [8] and a 3-μm-thick lightly Si-doped n$^-$-GaN layer (n ≈ $1.7 \times 10^{16}$ cm$^{-3}$) for sample B. These 3-μm-thick u-GaN/n$^-$-GaN layers were used as the absorption layers to generate holes and electrons.

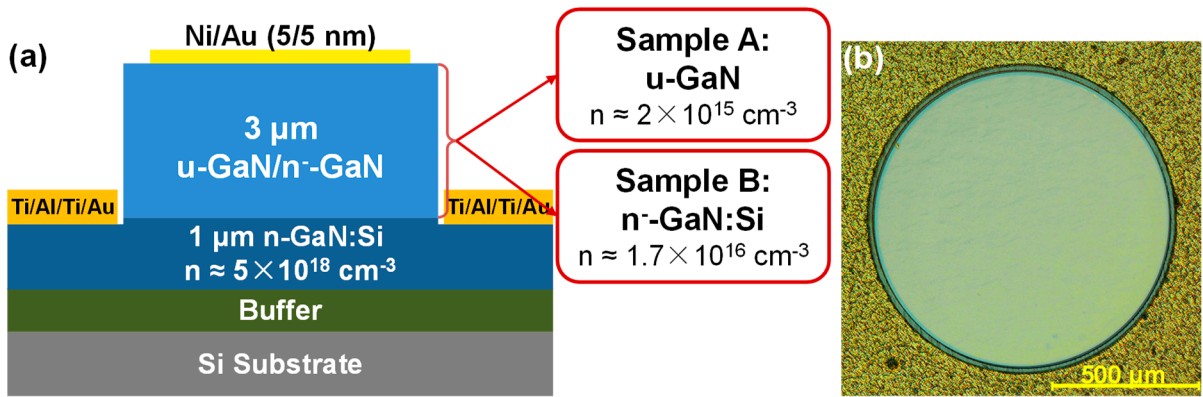

**Figure 1.** (**a**) Cross-sectional sketch and (**b**) top-view microscopy image of one as-fabricated GaN-on-Si Schottky barrier UV PD.

After the development of the materials, both samples were fabricated into PD devices by using photolithography technology and inductively coupled plasma (ICP) etching. The predominant transparent Schottky contact was formed on the u-GaN/n$^-$-GaN mesa by using a Ni/Au (5/5 nm) metal stack, followed by annealing under $N_2$ ambience at 600 °C for 300 s. The Ti/Al/Ti/Au (20/150/50/100 nm) metal stack was deposited on the exposed n-GaN and annealed under $N_2$ ambience to form good ohmic contact. Both contact metals were deposited by magnetron sputtering and annealed using the rapid thermal annealing facility. Figure 1b shows a top-view microscopy image of one circular Schottky barrier UV PD with a radius of 1000 μm.

The cathodoluminescence (CL) technique was used to characterize the material quality of the absorption layer [9], and the current–voltage (I-V) characteristics were measured in the dark using the Keithley 4200-SCS/F parameter analyzer. The responsivities of the fabricated samples were measured by a home-made spectral response measurement system composed of a Xenon lamp, monochromator, chopper and lock-in amplifier. In order to characterize the response time, a 365-nm UV LED bulb with a peak light intensity of 280 mW/cm$^2$ driven by a pulsed power source was used to produce pulsed light signals, and the fabricated samples illuminated by the pulsed light signals were connected with an oscilloscope to output the photo response signal; thus, the response time characteristics could be determined using the time resolution function of the oscilloscope. All the measurements were carried out at room temperature.

## 3. Results and Discussion

Figure 2 shows the CL images of samples A and B. In the images, the dark spots indicate a lack of emission intensity, which occurs mainly because most of the carriers recombine non-radiatively due to the TDs. Therefore, the dark spots can be treated as indicators of TDs. By counting the dark spots, the TD density of sample A was calculated to be approximately $2.0 \times 10^8$ cm$^{-2}$, which is almost equal to that of sample B ($2.4 \times 10^8$ cm$^{-2}$). Therefore, it can be concluded that light Si doping does not increase the TD density of GaN. It is noted that the TD density of GaN grown on the Si substrate is considerably lower compared with most of the reported results in the literature [10,11].

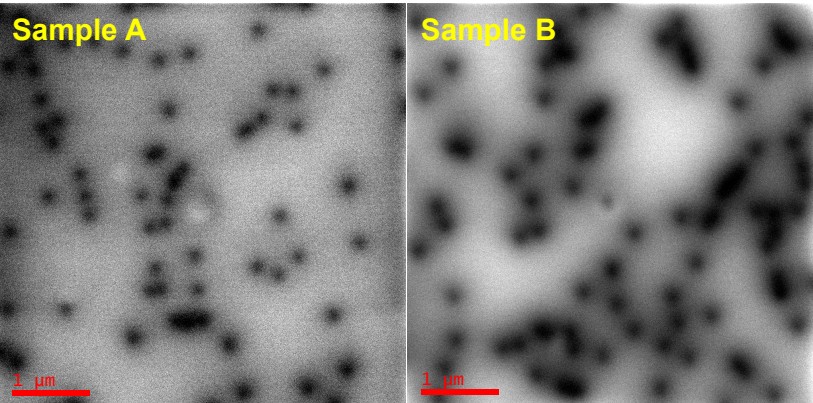

**Figure 2.** The CL images of samples A and B.

Figure 3 shows the normalized CL spectra of samples A and B. Both samples have three emission peaks, including a band-edge emission peak at approximately 366 nm, a blue luminescence (BL) band at around 430 nm and a typical yellow luminescence (YL) band at around 560 nm, which are often observed in GaN produced by MOCVD [12,13]. The BL band in GaN might be due to transitions from the conduction band or a shallow donor level to a relatively deep acceptor level [13–15], and the YL band might be attributed to the optical transition from the native donor level to the carbon-related acceptor level [13,16]. As shown in Figure 3, for both samples A and B, the intensities of the defect-related emission are almost the same and are fairly weak compared with those of the band-edge emission, which indicates that light Si doping does not significantly increase the point defects of GaN, and both samples A and B display a high-quality absorption layer.

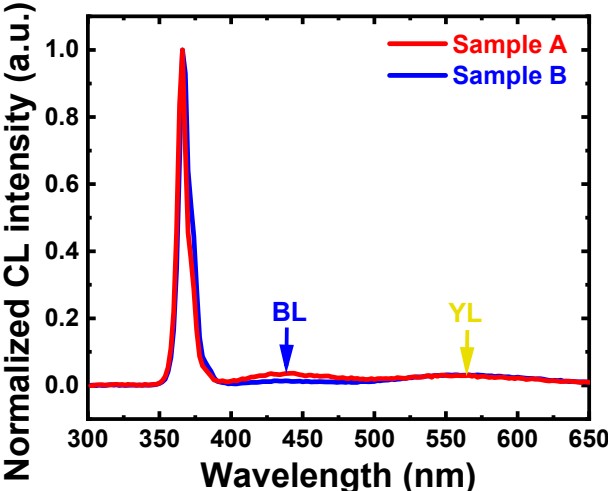

**Figure 3.** Normalized CL spectra of samples A and B.

Figure 4a shows the I-V curves of samples A and B, measured in the dark. The reverse leakage current of sample B is much larger than that of sample A, which is mainly due to the light Si doping. The Schottky barrier height (SBH) of samples A and B was calculated as 0.82 and 0.67 eV under dark conditions and as 0.79 and 0.65 eV under illumination, and the ideality factors were 2.4 and 2.3, respectively. It was confirmed that a higher doping concentration would result in a larger built-in electric field and a much narrower depletion region; thus, the SBH would be lowered by the image force effect and the tunneling probability might be increased [17,18]. Moreover, the conductivity of n⁻-GaN is higher than that of u-GaN, contributing to the large reverse leakage current. In addition, the interface states might also partially contribute to the leakage current in Ni/n-GaN Schottky contact. Guo et al. inferred that a series of compounds formed at the interface after annealing contribute to the leakage current [19]. The slight reduction in SBH under illumination might be due to photo-generated carriers captured by traps at the interface of Schottky contact.

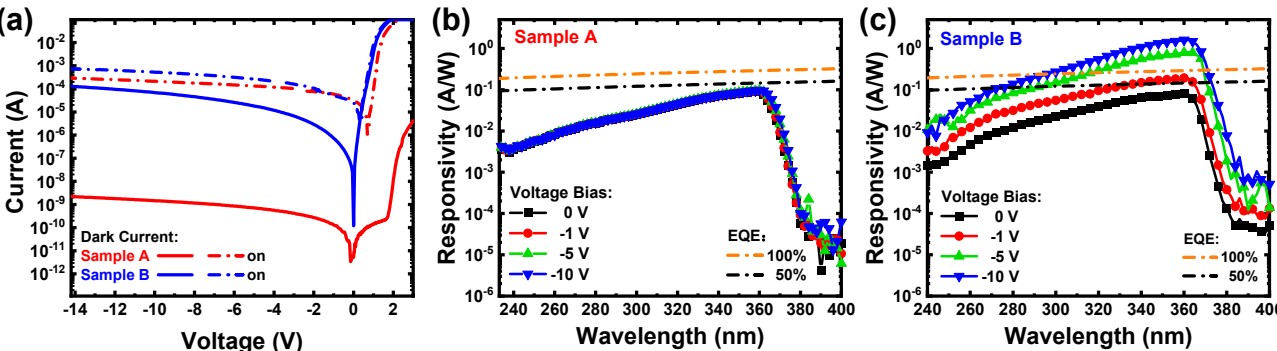

**Figure 4.** (**a**) I-V curves of samples A and B measured in dark; the spectral response curves of samples (**b**) A and (**c**) B under various reverse biases.

Figure 4b,c show the spectral response curves of samples A and B. Under zero bias, the responsivities of samples A and B are almost the same at approximately 0.09 A/W at 360 nm. When the reverse bias is increased, the responsivity of sample A is almost unchanged. However, for sample B, the responsivity increases significantly with the reverse bias, reaching 1.6 A/W at −10 V. According to the literature, this notable increase might be ascribed to the photoconductive gain induced by the persistent photoconductivity (PPC) [20,21]. The mechanism of the photoconductive gain and PPC phenomenon will be discussed later.

We measured the response characteristics of both samples. Figure 5a,b show the time-dependent photo response curves (photovoltaic) of samples A and B. Sample A exhibits rapid rise and decay at various reverse biases. However, sample B shows slow rise and decay, with an apparent persistent photocurrent under −3 and −5 V. Figure 5c shows the summarized rise and decay times of samples A and B under various reverse biases. The rise and decay times were defined as the time required for the photo response signal to increase from 10% to 90% and to decrease from 90% to 10% of the maximum value, respectively [22]. For sample A, the rise time increased and the decay time decreased slightly as the reverse bias was increased. Both the rise and decay times were very short. However, for sample B, the rise and decay times were very low under zero bias and increased rapidly along with the increase in reverse bias.

For sample A, the variation in the rise and decay times might indicate that carriers are captured and recombined faster by the shallow traps under a higher electric field [23]. When the reverse bias is increased, the generation rate of photo-generated carriers ($v_1$) is almost constant under a certain illumination. However, the capture and recombination rate ($v_2$) increases. Due to the competitive relationship between the generation and capture processes, the rise time and decay time are proportional to $1/|v_1 - v_2|$ and $1/|v_2|$,

respectively. Hence, the rise time increases and the decay time reduces with increasing reverse bias, as shown in Figure 5c. For sample B, according to the literature, the large gain and long response time might be ascribed to the PPC effect, which is mainly caused by the reduction in SBH [20,23]. According to the literature, the doped Si atoms can be treated as positively charged centers; these charging centers and related defects can capture or release the carriers under a higher electric field [13,24]. Under illumination, these traps might be excited massively under a high electric field; thus, the SBH will be lowered and the capture and recombination rate ($v_2$) of photo-generated carriers will be increased under higher reverse bias, greatly increasing the photocurrent and the time of the photocurrent rising process. When illumination is removed, these traps and SBH will be slowly recovered because the carriers drift away gradually due to the electric field. As more deep traps are generated in higher electric field, the photocurrent and the response time increase with the increasing reverse bias, as shown in Figure 5c.

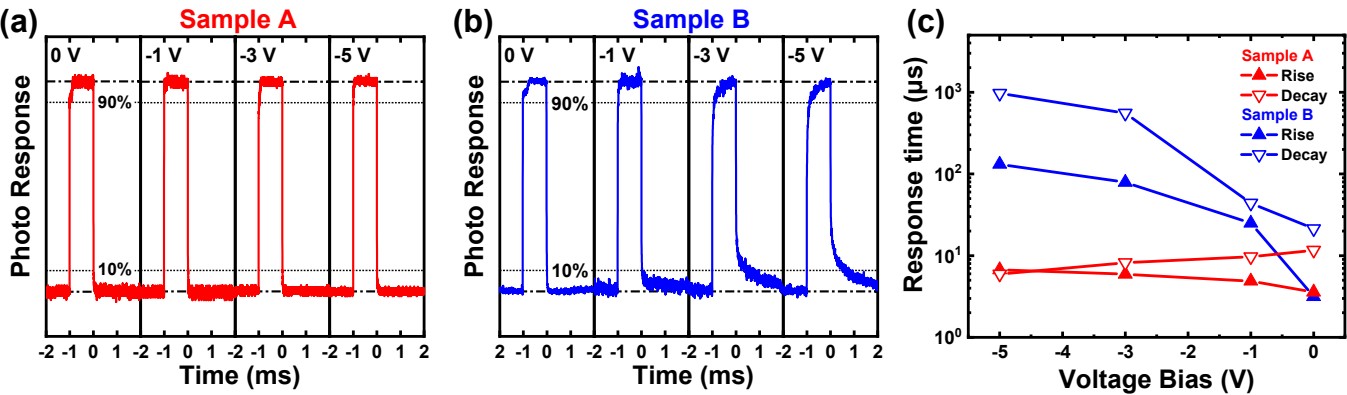

**Figure 5.** Time-dependent photo response curves (photovoltaic) of samples (**a**) A and (**b**) B at various reverse biases; (**c**) the rise and decay times of samples A and B at various reverse biases. The pulse frequency is 200 Hz and the duty cycle is 20%.

## 4. Conclusions

In summary, the effect of Si doping on the performance of a GaN Schottky barrier ultraviolet photodetector grown on Si was studied. It was found that light Si doping does not increase the threading dislocations and point defects in the material, but it greatly affects the device's performance. The lightly Si-doped device exhibited an obvious gain and high responsivity at reverse bias voltage, even reaching 1.6 A/W at −10 V, which was mainly due to the significant persistent photoconductivity confirmed in the measurement of the time-dependent photo response. This kind of device can be applied in areas that require high responsivity but less demanding response time. Meanwhile, the unintentionally doped device showed a small leakage current and fast response time and is therefore better suited to high-speed UV detection.

**Author Contributions:** F.L., W.C. and M.F. performed experiment and data processing; F.L., W.C. and M.F. performed data analysis; Y.H., J.L., X.S. and X.Z. provided material; Q.S., Q.W. and H.Y. supervised the whole study. All authors have read and agreed to the published version of the manuscript.

**Funding:** This research was funded by Guangdong Province Key-Area Research and Development Program (Grant Nos. 2019B010130001, 2019B090904002, 2019B090909004, 2019B090917005, and 2020B010174004), the Natural Science Foundation of China (Grant Nos. 61775230, 61804162, 61874131, and 62074158), the Strategic Priority Research Program of CAS (Grant Nos. XDB43000000 and XDB43020200), the Key Research Program of Frontier Sciences, CAS (Grant Nos. QYZDB-SSW-JSC014 and ZDBS-LY-JSC040), the Jiangsu Provincial Key Research and Development Program (BE2020004-2), the Natural Science Foundation of Jiangsu Province (Grant No. BK20180253), and the Suzhou Science and Technology Program (Grant No. SYG201927).

**Acknowledgments:** We are thankful for the technical support from Nano Fabrication Facility, Platform for Characterization and Test, and Nano-X of SINANO, CAS.

**Conflicts of Interest:** The authors declare no conflict of interest.

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
