# Peer review of "Effect of Si Doping on the Performance of GaN Schottky Barrier Ultraviolet Photodetector Grown on Si Substrate"

_photonics, doi:10.3390/photonics8020028_

Round 1

Reviewer 1 Report

In this paper, the authors report on the UV detectors based on GaN grown on Si substrate. They also studied the Si doping effect on the photoresponse properties. It was found the photoresponse properties could be controlled by the dopant concentration. The results are very interesting and I have some comments as follows on this paper.

(1) Please show in the text the UV light intensity and the chopping frequency.

(2) Please analyze the SBH and ideality factor of the diodes from sample A and B.

(3) Especially, please show the I-V curves under UV light illumination and analyze the SBH height by comparing that under dark condition. 

(4) The authors claimed PPC. Although there is, it will be better to show the time response without chopping or smaller chopping frequency.  

Reviewer 2 Report

The submitted paper Effect of Si doping on the performance of GaN Schottky barrier ultraviolet photodetector grown on Si Substrate  deals with characterization of GaN based UV detectors. Paper is well organised, however before the publication some points has to be addressed:

  • line 53 - subscript of 3 in NH3 is missing
  • 2. Materials and Methods - how author determine the carrier concentration of epitaxial structure A and B? Why they not shown measurements results of it?
  • line 74 - ... absorption layer,8 and the current... - some writing error
  • Figure 3 - I recommend to add Figure 3b where only BL and YL will be shown (for λ>400 nm)
  • lines 111-120 - in general the dark current is proportional to carrier concentration to power of 2, so the authors remarks are correct
  • Authors compare both photodetectors and conclude that different behaviour of rise and decay times are related with traps. Did authors perform some DLTS or even optical DLTS measurements of sample A and B to estimate the defect concentration and their localization with respect to conduction and valence band edges? In my opinion such investigations will significantly enhance the understanding of improvement of silicon doped device performance.

Round 2

Reviewer 2 Report

I would like to thank authors for their response on my remarks. I am satisfied with the revision and I recommend paper for publication.